# RSC primes the quiescent genome for hypertranscription upon cell-cycle re-entry

Christine E Cucinotta, Rachel H Dell, Keean CA Braceros, Toshio Tsukiyama*

Basic Sciences Division, Fred Hutchinson Cancer Research Center, Seattle, United States

**Abstract** Quiescence is a reversible $G_0$ state essential for differentiation, regeneration, stem-cell renewal, and immune cell activation. Necessary for long-term survival, quiescent chromatin is compact, hypoacetylated, and transcriptionally inactive. How transcription activates upon cell-cycle re-entry is undefined. Here we report robust, widespread transcription within the first minutes of quiescence exit. During quiescence, the chromatin-remodeling enzyme RSC was already bound to the genes induced upon quiescence exit. RSC depletion caused severe quiescence exit defects: a global decrease in RNA polymerase II (Pol II) loading, Pol II accumulation at transcription start sites, initiation from ectopic upstream loci, and aberrant antisense transcription. These phenomena were due to a combination of highly robust Pol II transcription and severe chromatin defects in the promoter regions and gene bodies. Together, these results uncovered multiple mechanisms by which RSC facilitates initiation and maintenance of large-scale, rapid gene expression despite a globally repressive chromatin state.

## Introduction

For decades, scientists have used budding yeast to uncover mechanisms of chromatin regulation of gene expression, and the vast majority of these studies were performed in exponentially growing (hereafter log) cultures (*Rando and Winston, 2012*). Log phase, however, is not a common growth stage in unicellular organism lifecycles. Furthermore, many cell populations in multicellular organisms, such as in humans, are not actively dividing (*Sagot and Laporte, 2019*; *Rittershaus et al., 2013*; *Cheung and Rando, 2013*). Indeed, the majority of 'healthy' cells on earth are not sustained in a persistently dividing state (*Rittershaus et al., 2013*). Non-proliferating cells reside in a $G_0$ state, which generally means that these cells are either terminally differentiated, senescent, or quiescent. The quiescent state provides advantages to organisms: quiescence allows cells to remain dormant for long periods of time to survive harsh conditions or to prevent over-proliferation (*Rittershaus et al., 2013*; *Cheung and Rando, 2013*; *Tümpel and Rudolph, 2019*; *Sagot and Laporte, 2019*). Notwithstanding this so-called 'dormant state', quiescent cells can exit quiescence and re-enter the mitotic cell-cycle in response to growth cues or environmental stimuli, which distinguishes quiescence from other $G_0$ states. A major hallmark of quiescence is the chromatin landscape – vast histone de-acetylation and chromatin compaction occur during quiescence entry (*McKnight et al., 2015*; *Young et al., 2017*; *Swygert et al., 2019*). These events happen alongside a global narrowing of nucleosome-depleted regions (NDRs) and increased resistance to micrococcal nuclease (MNase) digestion, indicating a repressive chromatin environment (*McKnight et al., 2015*). Together, these features of quiescent cells point to a critical role for chromatin regulation of the quiescent state. However, the role of chromatin regulation upon exit from quiescence is unknown.

Reversibility is a conserved hallmark of quiescent cells and is required for proper stem-cell niche maintenance, T-cell activation, and wound healing in metazoans (*Cheung and Rando, 2013*; *Chapman et al., 2020*). We sought to elucidate molecular mechanisms by which cells can overcome this repressive chromatin environment to re-enter the mitotic cell cycle. Given its genetic tractability,

**\*For correspondence:**
ttsukiya@fredhutch.org

**Competing interests:** The authors declare that no competing interests exist.

the ease by which quiescent cells can be purified, and high level of conservation among chromatin and transcription machinery, we turned to the budding yeast, *Saccharomyces cerevisiae* (*Gray et al., 2004*). We can easily isolate quiescent yeast cells after 7 days of growth and density-gradient centrifugation. In this context, we can study pure populations of quiescent yeast, a cell fate that is distinct from other cell types present in a saturated culture (*Allen et al., 2006*).

Since DNA is wrapped around an octamer of histone proteins in increments of ~147 bp to form nucleosomes (*Luger et al., 1997*), enzymes must move nucleosomes to give access to transcription initiation factors (*Lorch et al., 1987*). One such enzyme is the SWI/SNF family member, RSC, which is a 17-subunit chromatin-remodeling enzyme complex (*Hainer and Kaplan, 2020*). RSC contains an ATP-dependent translocase, Sth1 (*Cairns et al., 1996*; *Du et al., 1998*; *Saha et al., 2002*; *Zofall et al., 2006*), multiple subunits with bromodomains (more than half of all bromodomains in the yeast genome are in RSC) and two zinc-finger DNA-binding domains, which allow RSC to target and remodel chromatin (*Kasten et al., 2004*; *Angus-Hill et al., 2001*). Many components of the RSC complex are essential for viability in budding yeast and the complex is conserved in humans, where it is named PBAF. In humans, mutations in PBAF genes are associated with 40% of kidney cancers (*Varela et al., 2011*), and 20% of all human cancers contain mutations within SWI/SNF family genes (*Kadoch et al., 2013*), underscoring the importance of such complexes in human health.

The best-described role for RSC in regulating chromatin architecture and transcription is its ability to generate NDRs, by sliding or evicting nucleosomes (*Badis et al., 2008*; *Hartley and Madhani, 2009*; *Prajapati et al., 2020*). Moving the +1 nucleosome allows for TATA binding protein (TBP) promoter binding and transcription initiation (*Kubik et al., 2018*). To this end, RSC mostly localizes to the −1, +1, and +2 nucleosomes in log cells (*Ng et al., 2002*; *Yen et al., 2012*; *Ramachandran et al., 2015*). However, RSC has also been implicated in the transcription elongation step where it tethers to RNA polymerase and can localize to gene bodies (*Soutourina et al., 2006*; *Spain et al., 2014*; *Biernat et al., 2021*). Additionally, RSC binds nucleosomes within the so-called 'wide NDRs', where there are MNase-sensitive nucleosome-sized fragments, known as 'fragile' nucleosomes (*Xi et al., 2011*; *Knight et al., 2014*; *Teif et al., 2014*; *Kubik et al., 2015*). These RSC-bound nucleosomes are likely partially unwrapped to aid in rapid gene induction (*Kubik et al., 2015*; *Floer et al., 2010*; *Brahma and Henikoff, 2019*; *Schlichter et al., 2020*).

In this study, we investigated how genes are transcribed during the first minutes of quiescence exit. We were particularly interested in uncovering mechanisms to overcome highly repressive chromatin found in quiescent cells. Unexpectedly, ~50% of the yeast genome was transcribed by RNA polymerase II (Pol II) by the first 10 min of exit, despite the highly repressive chromatin architecture present in quiescence. We found that this hypertranscription (*Percharde et al., 2017*) event is RSC dependent and that RSC binds across the genome to ~80% of NDRs in quiescent cells. Upon RSC depletion, we observed canonical abrogation of transcription initiation, defects in Pol II clearance past the +1 nucleosome, and gross Pol II mislocalization, resulting in abnormal upstream initiation and aberrant non-coding antisense transcripts. We further showed that RSC alters chromatin structure to facilitate these processes. Taken together, we propose a model in which RSC is bound to NDRs in quiescent cells to facilitate robust and accurate burst of transcription upon quiescent exit through multiple mechanisms.

## Results

### Hypertranscription occurs within minutes of nutrient repletion post-quiescence

To determine the earliest time at which transcription reactivates during quiescence exit, we fed purified quiescent cells YPD medium and took time points to determine the kinetics of Pol II C-terminal domain (CTD) phosphorylation by western blot analysis (*Figure 1A*). Unexpectedly, Pol II CTD phosphorylation occurred within 3 min (*Figure 1A*, compare lanes 1 and 2), which was our physical limit of isolating cells during this time course. To determine which transcripts were generated during these early quiescence exit events, we performed nascent RNA-seq using 4-thiouracil (4tU) to metabolically label new transcripts (*Miller et al., 2011*; *Duffy et al., 2015*). In agreement with the western blot analysis, we observed a high level of transcriptional activation within a few minutes of nutrient repletion (*Figure 1B*). Based on our western blot result, the highest Pol II CTD

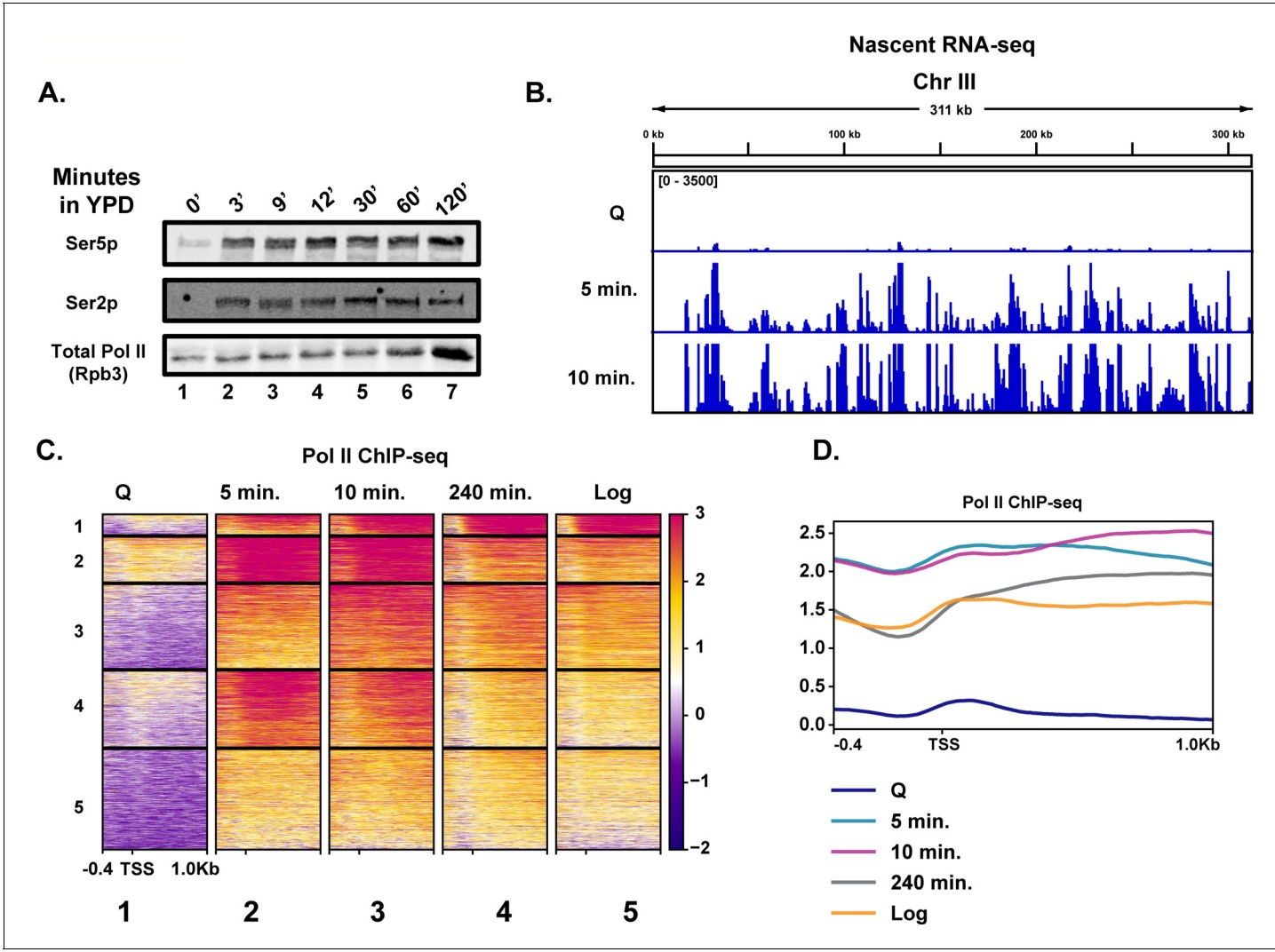

**Figure 1.** Rapid hypertranscription occurs upon nutrient repletion of quiescent cells. (A) Western blots were probed with antibodies to detect Ser5p and Ser2p of the CTD of Rpb1 subunit of Pol II. An antibody against the Rpb3 subunit of Pol II was used as a loading control. (B) Nascent RNA-seq analysis. (C) Pol II ChIP-seq analysis. Heatmaps show k-means clusters of 6030 genes. Genes are linked across the heatmaps. (D) Metaplots of ChIP-seq data shown in (C) without k-means clustering.

The online version of this article includes the following source data and figure supplement(s) for figure 1:

**Source data 1.** Gene Ontology analysis of genes transcribed during quiescence exit.

**Figure supplement 1.** Rapid and broad transcription occurs during the first minutes of exit.

**Figure supplement 1—source data 1.** Differential expression analysis of genes expressed in exit.

**Figure supplement 2.** Genes enriched in each cluster of genes shown in *Figure 1C*.

phosphorylation is observed ~10 min after refeeding. Consistent with this result, we observed the highest level of nascent transcripts at the 10 min time point, where 3202 RNAs (~50% of annotated loci) were statistically significantly increased by twofold compared to the 0 min time point (*Figure 1B*, *Figure 1—figure supplement 1A*). Given how quickly Pol II was phosphorylated and transcripts were generated, we sought to determine whether high levels of Pol II were already bound to the early exit genes in the quiescent state, as was observed previously in a heterogenous population of stationary phase cells (*Radonjic et al., 2005*). To this end, we performed spike-in-normalized ChIP-seq analysis of Pol II in quiescent cells and at several time points following refeeding (*Figure 1C*, *Figure 1—figure supplement 1B*). Low Pol II occupancy levels (compare heatmaps 1 and 5) were detected in quiescent cells, which agrees with our western blot and RNA-seq analyses and previously published literature (*McKnight et al., 2015*; *Young et al., 2017*; *Swygert et al.,*

2019). This implied that Pol II is not paused (*Figure 1C*, compare heatmaps 1 and 2) in quiescent cells and suggested that Pol II needs to be recruited de novo for rapid initiation and elongation. In support of this conclusion, we detected only low levels of the pre-initiation complex subunit TFIIB bound to genes in quiescent cells, which increased approximately threefold by 5 min of exit (*Figure 1—figure supplement 1C*), despite no changes in the abundance of the protein (*Figure 1—figure supplement 1D*).

Highlighting the high level of transcription occurring in the first 10 min of quiescence exit, we observed a drop-off in Pol II occupancy levels around the first G2/M phase (240 min) (*Figure 1C,D*, *Figure 1—figure supplement 1E*). Indeed, when the data were sorted into k-means clusters across the time course, we noticed that many of the genes expressed in the 240 min time point were similar, but still not identical, to those expressed in log cells, suggesting a recovery to log-like gene expression profile takes hours post refeeding (*Figure 1C*, compare columns 4 and 5, *Figure 1D*). There was a ~1.7-fold increase in overall Pol II occupancy in the 10 min time point relative to that of log cells (*Figure 1D*, *Figure 1—figure supplement 1B*). Genes within each cluster had some enriched Gene Ontology (GO) terms, particularly in cluster 1, where rRNA processing and translation-associated genes were well represented (*Figure 1—figure supplement 2*). Together, these results demonstrate that transcription activates extremely rapidly and robustly in response to nutrient repletion.

## Chromatin bears hallmarks of repression during early quiescent exit time points

Given the exceptionally high transcriptional response during the first 10 min of quiescence exit, we wondered whether chromatin changes reflected hypertranscription. To this end, we performed ChIP-seq analysis of H3 to measure nucleosome occupancy levels genome wide over time. Global H3 patterns during the early exit time points, especially at the 5 min time point, were more similar to that of the quiescent state than to the 240 min time point (*Figure 2A*, compare columns 1–3), despite higher transcription levels. The most striking changes in histone occupancy during the early time points were within NDRs, where the pattern at the 10 min time point resembles the 240 min time point (*Figure 2A,B*). However, the H3 profiles outside of NDRs (*Figure 2A*, compare column 1–3 and 4 to the right of NDR, and *Figure 2B*) remain similar to that of quiescent state during the early stage of quiescent exit. In addition to nucleosome occupancy, we tested nucleosome positioning using MNase-seq analysis where nucleosomes with 80% of the digested chromatin is represented by mononucleosomes. Globally, nucleosome positions were stable across the early exit time points (*Figure 2C*).

We next tested if a burst of histone acetylation occurred during these early exit time points to help overcome the repressive quiescent chromatin environment. To test this, we performed ChIP-seq analysis of H4ac using an antibody that recognizes penta-acetylated H4. Similar to nucleosome occupancy and positions, a modest increase in histone H4 acetylation occurred, but the levels did not reflect that of log cells (*Figure 2D,E*). This suggests that, while there was a strong transcriptional response during refeeding, histone acetylation was delayed. This is consistent with a previous study of a mixed population of saturated cultures where histone acetylation was found to occur later in exit (*Mews et al., 2014*). Together, our results are in agreement with a recent study demonstrating that histone acetylation takes place mostly as a consequence of transcription (*Martin et al., 2021*).

To assess a biological readout of the repressive chromatin environment, we turned to phenotypic analysis of TFIIS disruption. TFIIS is a general elongation factor that rescues stalled Pol II, and nucleosomal barriers have been shown to increase stalled Pol II (*Noe Gonzalez et al., 2021*). Given that Pol II stalling is common across the genome (*Churchman and Weissman, 2011*), it is paradoxical that the gene encoding TFIIS is not essential for viability in actively dividing cells, and its deletion does not cause strong growth defects (*Hubert et al., 1983*). Since Pol II must achieve a high level of transcription in the repressive chromatin environment during early quiescence exit, we hypothesized that TFIIS may play more critical roles during this period than during log culture. Indeed, in the absence of TFIIS (*dst1Δ*), quiescent yeast cells exhibited defects in cell-cycle re-entry, where cells lacking TFIIS stall at the first G1 during exit, which is not the case during the mitotic cell cycle (*Figure 2F*). These results collectively revealed that the chromatin environment remains repressive during early quiescence exit.

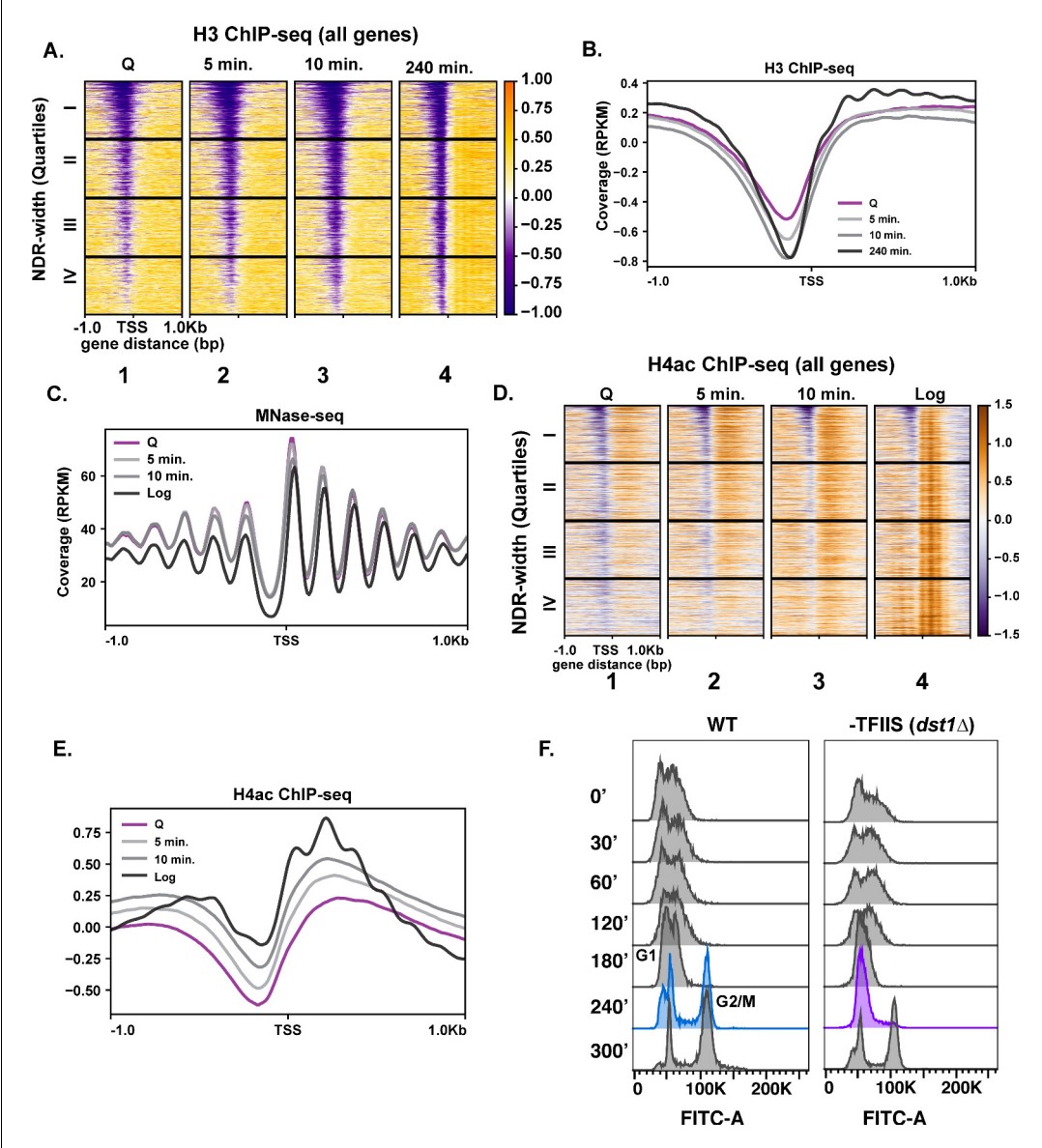

**Figure 2.** Repressive chromatin persists during early quiescence exit. (A, B) ChIP-seq of total H3 in quiescent cells and exit time points sorted into quartiles based on NDR width. (C) MNase-seq analysis of 6030 genes in Q (pink line), log (black line), and Q-exit time points 5 min (light gray line) and 10 min (dark gray line). (D, E) ChIP-seq analysis of penta-acetylated H4 (H4ac) in Q and log cells and exit time points. Genes are separated as in (B). (F) DNA content FACS analysis following Q exit in WT and a TFIIS-absent strain (*dst1Δ*).

## In quiescence, RSC re-localizes to NDRs of genes expressed in exit

Given the modest changes in chromatin at most genes during the early stage of quiescence exit (*Figure 2*), we wondered whether MNase-sensitive or 'fragile' nucleosomes were present at the promoters of rapidly induced genes in quiescence and were removed in early exit. Thus, we performed a weaker (low) MNase digestion (10% mononucleosomes) (*Figure 3A*) and compared it to the stronger (high) MNase digestion (80% mononucleosomes) (*Figure 3B*). Supporting our hypothesis, comparing the weaker MNase digest to the stronger MNase digest revealed that genes in the top two quarters of the NDR width have MNase-sensitive fragments in quiescent cells, which are reduced during exit (*Figure 3A*, *Figure 3—figure supplement 1A*). H3 occupancy levels as measured by ChIP-seq analysis were reduced across all four quartiles, with a greater change occurring in the top quartile (*Figure 3—figure supplement 1B*).

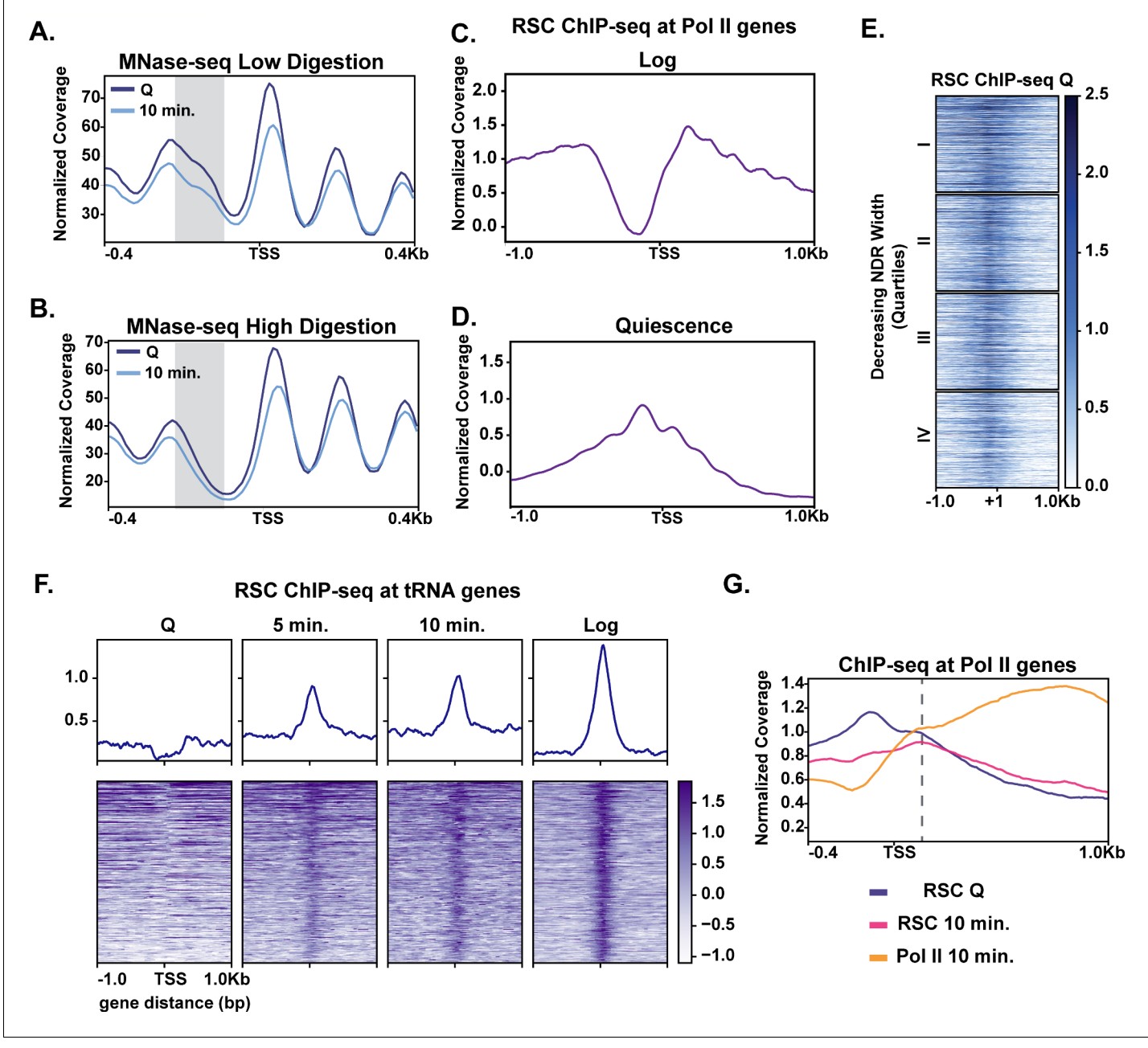

**Figure 3.** MNase sensitivity and quiescence-specific RSC re-localization indicate remodeling activity required for early exit. (A) MNase-digested chromatin to 10% mononucleosomes (low digestion). (B) Metaplot of MNase-digested chromatin to 80% mononucleosomes (high digestion) in Q and 10 min time points. (C, D) ChIP-seq of the catalytic RSC subunit in quiescent and log cells at Pol II-transcribed genes. (E) ChIP-seq analysis of RSC shown across quartiles based on MNase-seq determined NDR width. (F) ChIP-seq of RSC at tRNA genes. (G) ChIP-seq of RSC and Pol II comparing RSC movement with Pol II into gene bodies.

The online version of this article includes the following figure supplement(s) for figure 3:

**Figure supplement 1.** MNase-sensitive fragments and histone occupancy vary depending on genes transcribed in exit.

**Figure supplement 2.** Sth1 occupancy mirrors MNase-sensitive fragments.

**Figure supplement 3.** RSC binds NDRs upstream of genes turned on in exit.

It has been recently suggested that the ATP-dependent chromatin remodeler RSC can remove MNase-sensitive particles or fragile nucleosomes from promoters to activate transcription (*Kubik et al., 2018*). Additionally, it was proposed that RSC-bound nucleosomes are remodeling intermediates that render such nucleosomes more MNase sensitive (*Brahma and Henikoff, 2019*). Thus, RSC was a strong candidate for regulating rapid transcription activation during quiescence exit. We performed ChIP-seq analysis of the RSC catalytic subunit Sth1 in quiescent cells (*Figure 3E*, *Figure 3—figure supplement 2A*). In quiescence, Sth1 exhibited a striking difference in binding pattern compared to log cells (*Figure 3C,D*). Sth1 bound to the majority of NDRs at gene promoters in quiescent cells as judged by ChIP signal down the heatmap (*Figure 3E*, *Figure 3—figure supplement 3A*). This result was distinct from log cells, where RSC was reported to occupy the widest NDRs but otherwise bind the −1, +1, and +2 nucleosomes for most highly expressed genes (*Figure 3C*; *Yen et al., 2012*; *Kubik et al., 2018*; *Brahma and Henikoff, 2019*). Consistent with previous literature, the clusters containing more RSC ChIP signals also had MNase-sensitive fragments at NDRs (*Figure 3—figure supplement 2B*, cluster 1).

The RSC binding pattern in quiescent cells instead mirrored a recently described binding pattern in heat shock, where RSC and other transcription regulators *transiently* re-locate to the NDRs (*Vinayachandran et al., 2018*). In contrast to the heat shock response, however, we observed a stable, strong binding pattern of RSC in NDRs regardless of NDR width (*Figure 3E*). Another obvious distinction of RSC binding patterns between log and quiescence was observed at tRNA genes (*Figure 3F*). RSC's role in tRNA expression has been well studied in log cells (*Parnell et al., 2008*; *Mahapatra et al., 2011*; *Kumar and Bhargava, 2013*). In quiescence, RSC was occluded from tRNAs genes. Whereas upon exit, RSC rapidly targeted tRNAs, mimicking the log pattern. Together these data suggest that RSC adopts a quiescence-specific binding profile, one in which RSC is bound to NDRs broadly across the genome.

We next sought to gain insight into how quiescent RSC occupancy patterns might predict Pol II occupancy during exit. To this end, we compared localization of RSC and Pol II in quiescence and exit. We first found that the presence of RSC at NDRs in quiescent cells and strong transcription in exiting cells co-localized (*Figure 3—figure supplement 3A*). Next, we examined RSC occupancy changes during quiescence exit at Pol II-transcribed genes. During quiescence exit, RSC began to move out of NDRs and into gene bodies as transcription increased (*Figure 3G*). These results suggested that RSC facilitates transcriptional activation upon exit and raised the possibility that RSC binding in NDRs may be a mechanism for cells to prepare for quiescence exit.

## RSC depletion causes quiescent exit defects and global Pol II occupancy reduction during quiescence exit

To test the requirement of RSC in quiescence exit, we simultaneously depleted two essential subunits of the RSC complex, Sth1 and Sfh1, using the auxin degron system (*Nishimura and Kanemaki, 2014*), during quiescence entry (see Materials and methods; *Figure 3—figure supplement 3B*). Depletion of these subunits throughout the exit process (hereafter '-RSC') caused a dramatic defect in cell-cycle progression upon quiescence exit, where the cells exhibited strong delays in exiting the first G1 stage (*Figure 4A*). This result contrasted with that in cycling cells, where *rsc* mutants or conditional alleles cause G2/M arrest (*Tsuchiya et al., 1992*).

To determine the impact of RSC depletion on hypertranscription during quiescence exit, we performed Pol II ChIP-seq analysis on cells exiting quiescence. In the presence of RSC, Pol II levels peaked at 10 min and substantially decreased at 30 min after the exit (*Figure 4B*, compare columns 3 and 4). As is the case in log cultures (*Parnell et al., 2008*; *Kubik et al., 2019*; *Klein-Brill et al., 2019*), Pol II occupancy decreased in the absence of an intact RSC complex in Q cells and upon nutrient repletion thereafter (*Figure 4B*). Pol II occupancy did eventually increase over time in the RSC-depleted samples. However, even after 30 min, Pol II did not reach the peak level of occupancy seen at the 10 min mark in the +RSC condition (*Figure 4B*, compare heatmaps 3 and 8, and *Figure 4C*). This suggests that the defect in Pol II occupancy during quiescence exit was not solely due to slower kinetics during the initial exit stage.

As shown earlier in *Figure 3G*, we observed RSC leaving the NDRs and moving into gene bodies during quiescence exit. Therefore, we examined the impact of RSC depletion on nucleosome occupancy and positioning. H3 ChIP-seq showed that RSC is required for removal of histones within NDRs (*Figure 4D*), which is consistent with RSC's role as the 'NDR creator' (*Hartley and Madhani,*

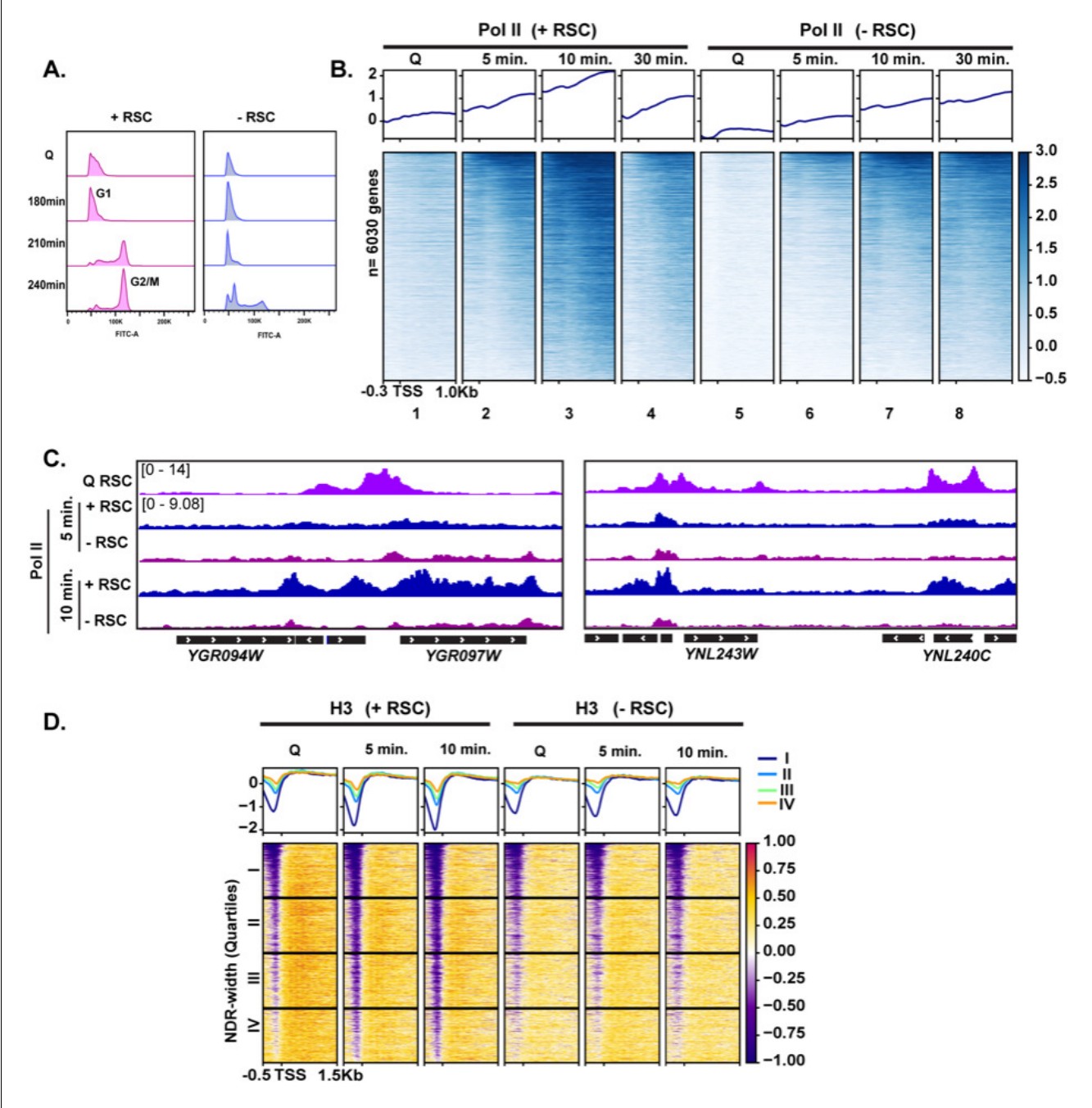

**Figure 4.** RSC is required for normal quiescence exit and hypertranscription upon nutrient repletion. (**A**) DNA content FACS analysis indicating cell-cycle progression during Q exit in the presence (+) or absence (−) of RSC. (**B**) ChIP-seq analysis of Pol II across time in the presence or absence of RSC. Genes are sorted in the same fashion for all heatmaps. (**C**) Example tracks of data shown in (**B**) with RSC ChIP-seq in Q cells added. (**D**) H3 ChIP-seq sorted by NDR width (as determined by MNase-seq experiments).

The online version of this article includes the following figure supplement(s) for figure 4:

**Figure supplement 1.** Comparison of different classes of genes transcribed in exit.

*2009*). We then plotted the data into the same k-means clusters shown in *Figure 1C* and cross compared TFIIB and RSC occupancy with RSC depletion on Pol II, nucleosome positions, and H3 occupancy at these sites (*Figure 4—figure supplement 1*). Genes across all clusters showed decreased Pol II occupancy, indicating Pol II loading defects shown in *Figure 4B*. However, genes that had high

TFIIB levels and were strongly expressed (clusters 1 and 2) still exhibited detectable Pol II occupancy when RSC was depleted (*Figure 4—figure supplement 1B*). This coincided with a reduction in MNase-sensitive nucleosomes even in the absence of RSC. While H3 levels increased at clusters 1 and 2, these genes had the lowest H3 occupancy even in the absence of RSC (*Figure 4—figure supplement 1C*). Together, these data suggest that chromatin regulation by RSC is the key contributor to Pol II occupancy defects during quiescence exit when RSC is depleted. We, however, note that transcriptional defects upon RSC depletion, rather than the loss of RSC itself, can be at least partly responsible for chromatin defects observed upon RSC depletion.

## RSC is required for Pol II passage through gene bodies

Given that RSC moves from NDRs into gene bodies during quiescence exit (*Figure 3G*), we next tested whether RSC could aid transcription after initiation. To this end, we selected ~2000 genes where RSC moved toward gene bodies and examined RSC localization at the 10 min time point of quiescent exit. This analysis showed uniform movement of RSC from NDR into gene bodies (*Figure 5A*). We next tested whether this RSC movement is dependent on Pol II transcription. To this end, we performed Sth1 ChIP-seq analyses during quiescence exit in the presence of a transcription inhibitor 1,10-phenanthroline (*Figure 5B*, Pol II control in *Figure 5—figure supplement 1A*). We once again utilized the clusters shown in *Figure 1C* to examine changes in localization at these sites. We note that at clusters 1 and 2, where Pol II normally is highly active, RSC is dramatically sequestered in the NDR (*Figure 5—figure supplement 1B*). This experiment demonstrated that the movement of RSC from NDRs into gene bodies was strongly inhibited by 1,10-phenanthroline, establishing that RSC re-localization during quiescent exit is dependent on Pol II transcription.

Co-transcriptional movement of RSC into gene bodies suggested a possibility that RSC may help Pol II passage through gene bodies. To test this, we determined the effects of RSC depletion on Pol II localization during early time points of quiescence exit. *Figure 5C,D* show that RSC depletion

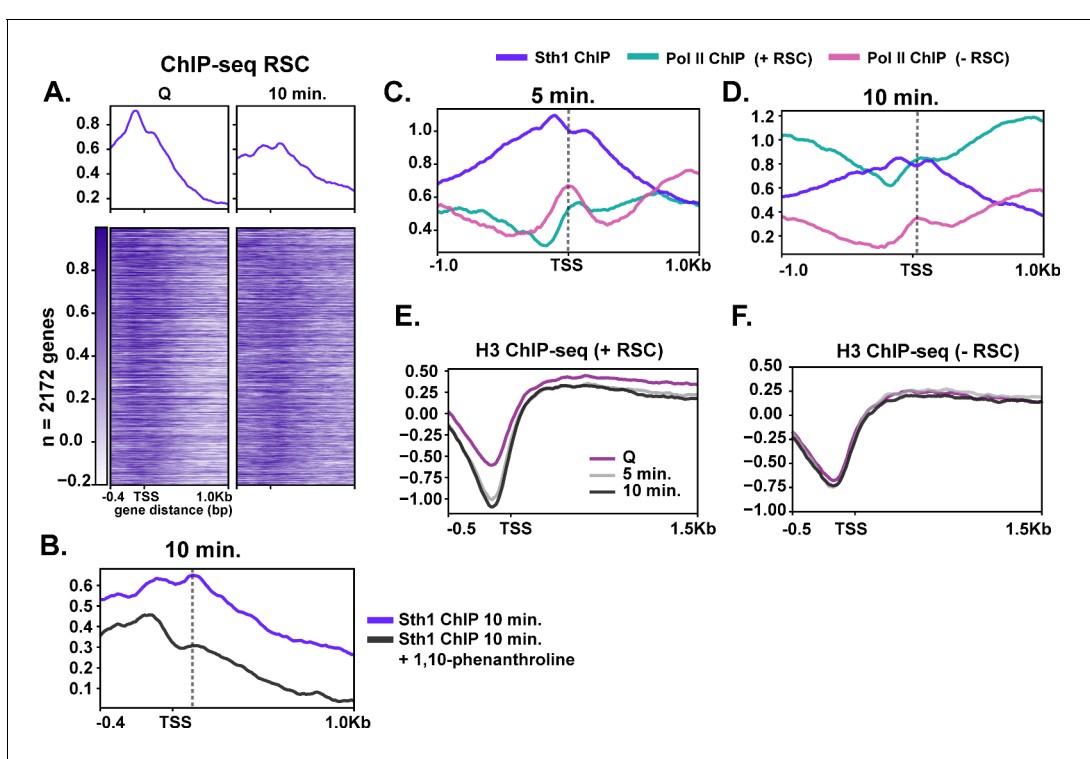

**Figure 5.** RSC depletion causes severe Pol II mislocalization defects during quiescence exit. (**A**) ChIP-seq of RSC in Q and 10 min time points. Genes are linked. (**B**) ChIP-seq of RSC at 10 min of exit in the presence and absence of the transcription inhibitor 1,10-phenanthroline. (**C, D**) ChIP-seq of RSC and Pol II during exit. (**E, F**) H3 ChIP-seq in quiescence and during exit in the presence and absence of RSC.

The online version of this article includes the following figure supplement(s) for figure 5:

**Figure supplement 1.** Inhibition of Pol II reduces Pol II occupancy and Sth1 movement into gene bodies.

affects Pol II localization in at least two ways during early quiescence exit. First, consistent with *Figure 4B*, the robust increase in the amount of Pol II over genes is strongly decreased upon RSC depletion. In addition, upon RSC depletion, Pol II sharply accumulates at transcription start sites (TSSs) at the 5 min mark, which continued to the 10 min mark. In sharp contrast, Pol II accumulates at slightly more downstream at the 5 min mark and moves mostly to downstream regions at the 10 min time point in the presence of RSC. At these loci, NDRs are relatively shallow in quiescence but histone density rapidly decreases upon quiescence exit in the presence of RSC (*Figure 5E*). In the absence of RSC at these sites, however, histone density is unexpectedly lower at NDRs in quiescence but does not change during quiescence exit (*Figure 5F*), suggesting defective chromatin structure at and downstream of the NDR. Together, these results are consistent with the notion that co-transcriptional movement of RSC facilitates passage of Pol II through nucleosomes immediately downstream of TSSs through chromatin regulation.

## RSC suppresses abnormal upstream transcription initiation

The fact that Pol II accumulated upstream of TSSs at the 5 min mark upon RSC depletion (*Figure 5C*) suggested possible defects in TSS selection. To test this possibility, we examined the 4tU-seq profiles in which there appeared to be an enrichment of RNA signal directly upstream and downstream of TSSs. We took the $\log_2$ ratio of RNA signal in the depleted condition versus the non-depleted condition at the 10 min time point. We sorted the genes using k-means clusters and found 864 targets in which upstream transcription was present (*Figure 6A*, three clusters shown in *Figure 6—figure supplement 1*, and an example of a representative locus in *Figure 6B*). At these sites, we observed RSC ChIP-seq signals at NDRs in quiescence and then spreading during exit (*Figure 6C*). Indeed, at *PTP3*, we observe opening of the NDR in the +RSC condition and the NDR remaining absent when RSC was depleted (*Figure 6B*).

This analysis revealed that upon RSC depletion, a large number of genes (864) exhibited increased nascent sense-strand RNA signals starting upstream of their normal TSSs, demonstrating widespread defects in TSS selection. Canonical NDRs at these sites were severely reduced in the absence of RSC (*Figure 6D*; *Figure 6—figure supplement 1D*). Examination of individual loci revealed that, in addition to filling of an NDR at the normal TSSs, an NDR is created upstream, which overlaps with ectopic transcription observed at an upstream TSS (see *Figure 6B* for an example). These results suggest that RSC facilitates selection of accurate transcription initiation sites through proper NDR formation upstream of protein coding genes during the burst of transcription during quiescence exit. This is likely a quiescence-specific function of RSC, or a result of the robust hyper-transcription event during exit, as depletion of Sth1 in cycling cells mostly repressed transcription initiation with relatively few new upstream TSSs (*Kubik et al., 2019*; *Klein-Brill et al., 2019*).

## RSC is required for suppression of antisense transcripts during quiescence exit

Given the robust transcriptional response during the early minutes of quiescence exit (*Figure 1*), we examined whether aberrant transcripts might also arise at RSC target loci during quiescence exit when RSC was depleted. We sorted the ratio of antisense transcript levels with and without RSC depletion into five k-means clusters (*Figure 7A*). We found antisense transcripts arising in the absence of RSC, particularly at clusters I and IV. RSC signals were observed at NDRs upstream of sense transcripts in all clusters, with cluster II having the lowest levels of RSC (*Figure 7B*) and the highest levels of sense transcription (*Figure 7A*). Most genes had RSC bound at the promoters of the sense genes in quiescence, with highest RSC binding in the cluster I genes (*Figure 7B*). Strikingly, nucleosome positioning and occupancy were heavily impacted in the cluster I and IV genes upon RSC depletion in the sense direction, where NDRs became more resistant to MNase and nucleosomes in gene bodies were shifted toward the 5'-ends of genes (*Figure 7C,D*). This was in contrast to genes in clusters II and V where NDRs were largely open (*Figure 7C,D*). These results collectively showed that chromatin structure at the cluster I and IV genes is especially dependent on RSC. In both clusters of genes, RSC signals and RSC-dependent chromatin changes are not apparent around the start sites of antisense transcripts. Therefore, suppression of antisense transcripts is unlikely to be a direct role for RSC. Instead, it is likely that these genes have an intrinsic property to allow

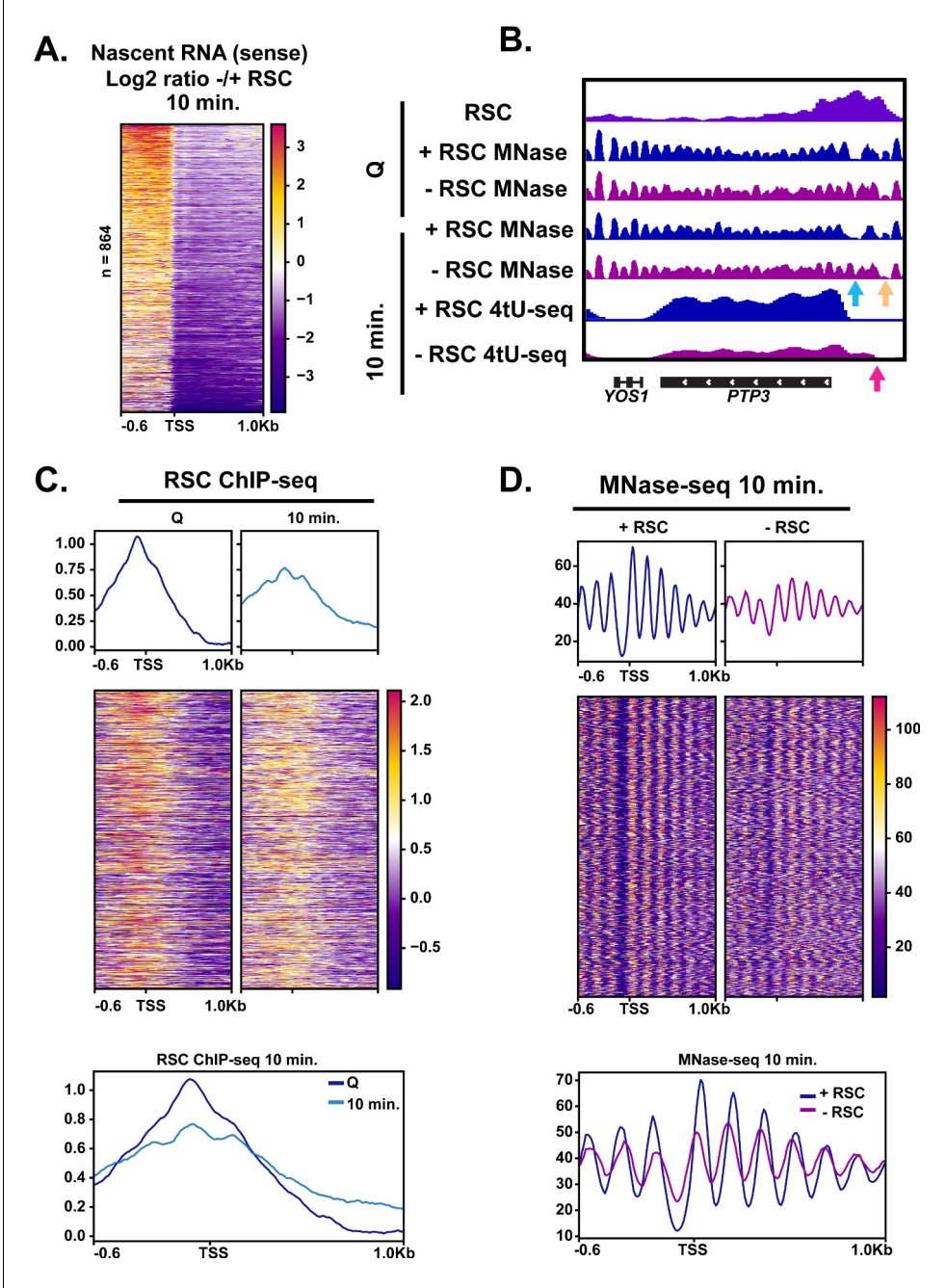

**Figure 6.** RSC depletion causes upstream transcription relative to canonical TSS. (A) Heatmap showing the $\log_2$ ratio of nascent sense transcripts in RSC-depleted versus non-depleted cells. Shown are 864 genes that have upregulated transcripts upstream of genes in the sense direction and have RSC ChIP signals. (B) Example gene of aberrant upstream transcript. Arrows direct to defects: blue arrow points to loss of NDR, yellow arrow points to gain of NDR, and pink arrow points to upstream RNA signal. (C) Heatmaps and metaplots of RSC ChIP-seq during Q and exit at genes shown in (A). (D) Heatmaps and metaplots of MNase-seq in exit at the genes shown in (A). The online version of this article includes the following source data and figure supplement(s) for figure 6:

**Source data 1.** List of genes with aberrant upstream start sites.
**Figure supplement 1.** Genes with cryptic upstream start sites when RSC is depleted have the most narrow NDRs in the absence of RSC.
**Figure supplement 1—source data 1.** List of genes in each cluster.

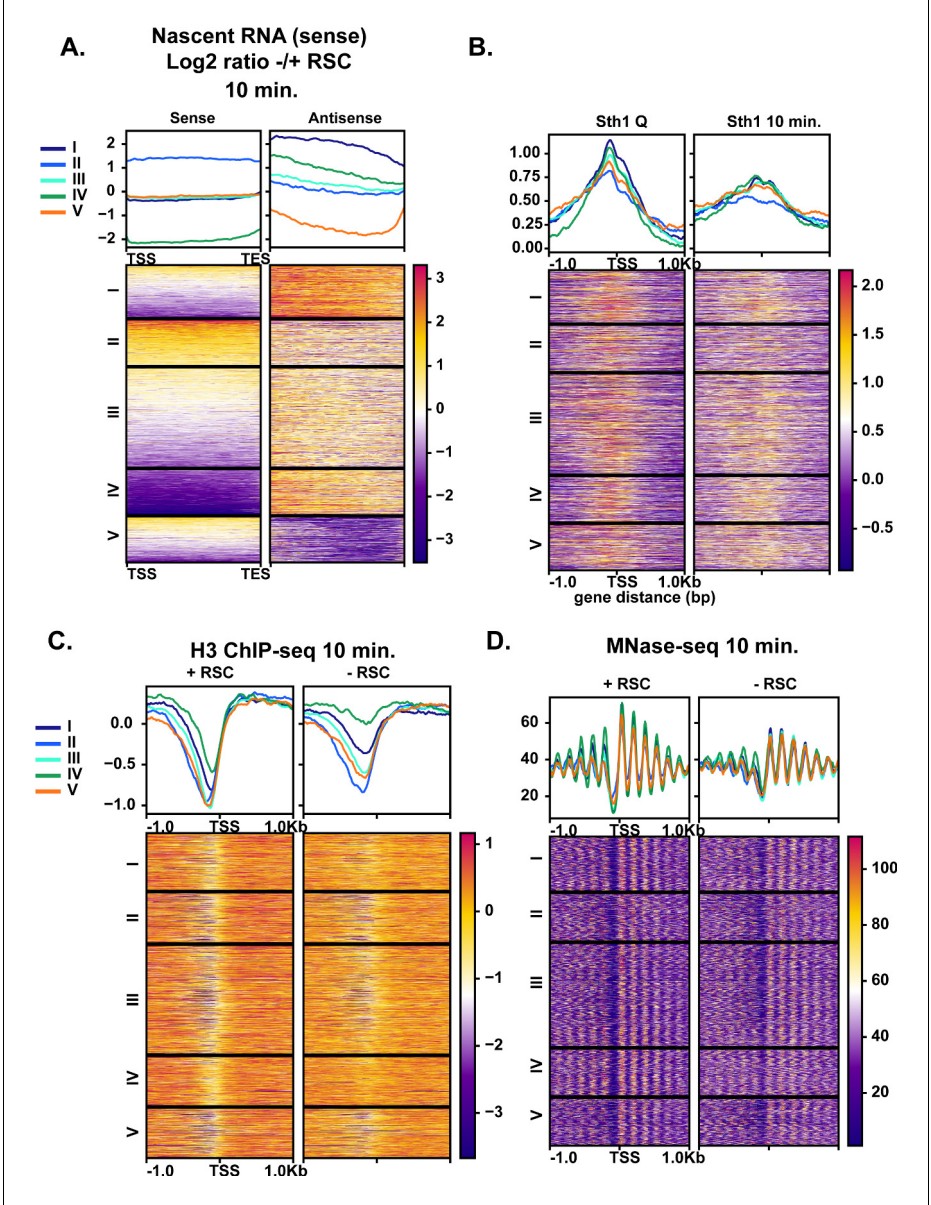

**Figure 7.** Aberrant antisense transcription arises when chromatin around sense transcripts is abrogated in the absence of RSC. (**A**) Heatmaps of the log$_2$ ratio of nascent RNAs that are RSC targets and give rise to antisense transcripts. Data are sorted into 5 k-means clusters based on the antisense transcripts. All data in this figure are sorted in the same fashion. (**B**) ChIP-seq of RSC in quiescent cells and during exit. (**C**) H3 ChIP-seq at the 10 min time point with and without RSC. (**D**) MNase-seq at the 10 min time point with and without RSC.

The online version of this article includes the following source data for figure 7:

**Source data 1.** List of genes in each cluster.

antisense transcription to occur when not properly regulated, and RSC is targeted to them to ensure sense transcription takes place through formation of proper NDRs.

## Discussion

In this report, we have shown that there is a rapid and robust transcriptional response during the very early minutes of quiescence exit (*Figure 8A*). This response is greatly dependent on the chromatin-remodeling enzyme RSC. We found that RSC promotes transcription at the right place and

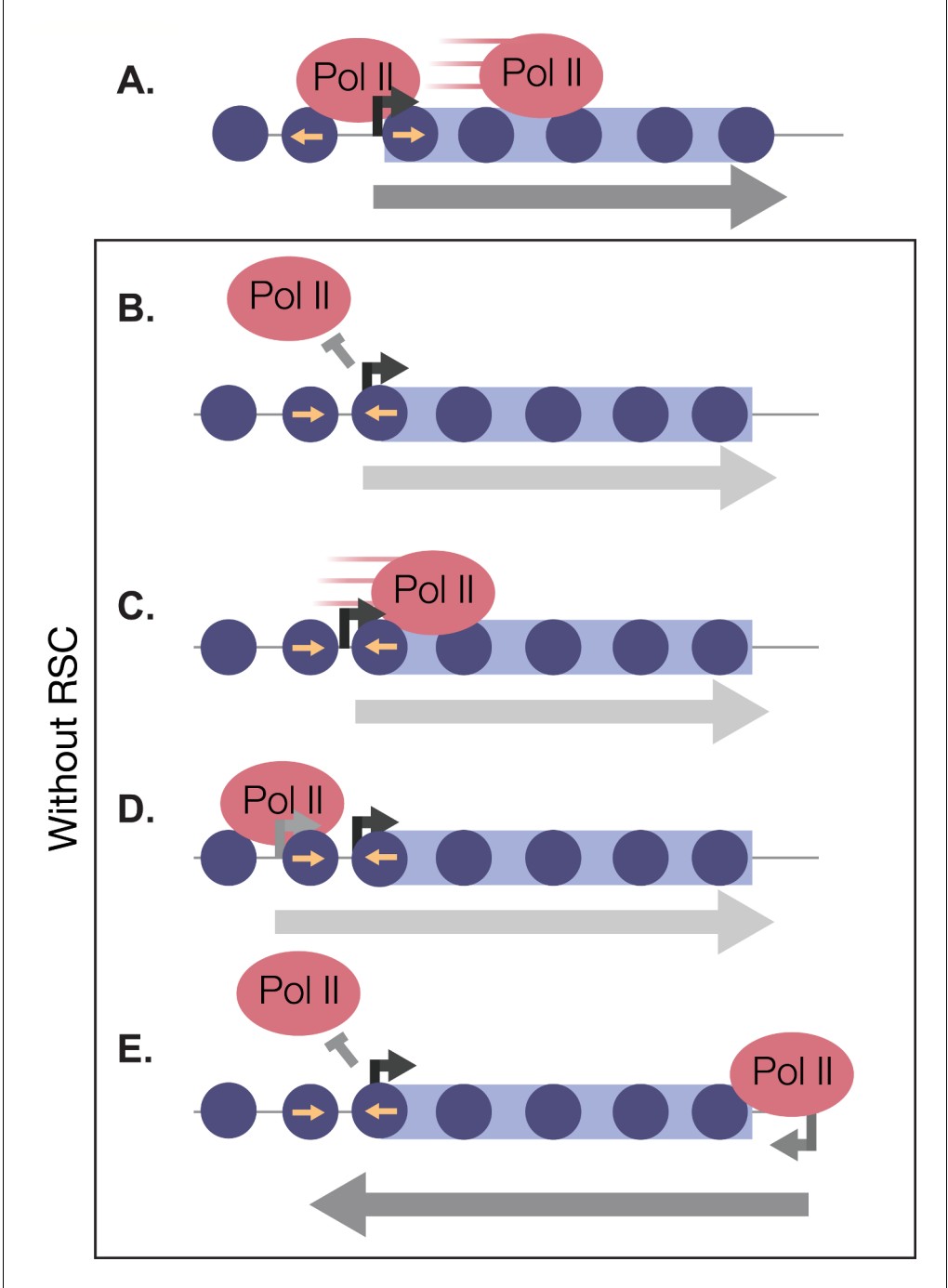

**Figure 8.** RSC bound to NDRs is poised in quiescence to safeguard the genome from aberrant transcription during quiescence exit. Upon quiescence exit, RSC shifts the +1 nucleosome to allow for Pol II occupancy and traverses into gene bodies (A). In the absence of RSC, NDRs are globally narrower and transcription initiation is blocked (B). At a subset of genes, RSC is required for efficient Pol II passage past the +1 nucleosome (C) and prevents upstream TSS selection (D). NDRs that are open despite RSC depletion become cryptic promoters and are utilized by transcription machinery to generate aberrant lncRNAs and antisense transcripts (E).

time in four different ways: (1) RSC promotes transcription initiation by creating NDRs in quiescence and maintaining them during exit (*Figure 8B*). (2) RSC moves into gene bodies and helps Pol II transcribe past the +1 nucleosome (*Figure 8C*). (3) RSC maintains proper NDR locations to allow for accurate TSS selection (*Figure 8D*). (4) RSC suppresses cryptic antisense transcription via generating

NDRs at the cognate sense genes (*Figure 8E*). Together, our results suggest that the massive transcriptional response requires highly accurate nucleosome positioning to allow for cells to exit from the quiescent state.

Quiescent yeast must downregulate their transcriptional program and generate a repressive chromatin environment in order to survive harsh conditions for extended time periods (*Gray et al., 2004*; *McKnight et al., 2015*; *Spain et al., 2018*; *Poramba-Liyanage et al., 2020*). How, then, do cells rapidly escape the quiescent state when conditions are favorable? In this study, we show that there is a broad and robust transcriptional response to nutrient repletion after quiescence, notwithstanding a relatively repressive chromatin environment that persists until the first G2/M phase after quiescence. Indeed, we identified a previously unknown phenotype for the deletion of the gene encoding yeast TFIIS, *dst1Δ*. High numbers of stalled Pol II are present in cycling cells (*Churchman and Weissman, 2011*) despite the little impact of deleting *DST1* on cycling cell growth. We speculate cells exiting quiescence may rely more heavily on TFIIS to transcribe through repressive chromatin (*Kireeva et al., 2005*; *Bondarenko et al., 2006*).

During quiescence, RSC re-locates to NDR upstream of Pol II-transcribed genes that are transcribed in exit. Although RSC binds and regulates chromatin around Pol III genes (*Ng et al., 2002*; *Parnell et al., 2008*), RSC is depleted at tRNA genes in quiescence and only returns during quiescence exit, further supporting the notion that RSC is globally re-targeted in quiescence. This is distinct from the transient NDR re-localization observed in heat shock (*Vinayachandran et al., 2018*), as what we observed in quiescence was a sustained and rather stable localization. How RSC binds to these new locations in quiescence is unknown. Given the distinct structure of quiescent chromatin there are several, non-mutually exclusive, explanations for RSC's binding pattern in quiescence. (1) The genome is hypoacetylated and thus RSC can no longer bind to acetylated nucleosomes in quiescence via its bromodomains (*Kasten et al., 2004*). However, given the highly robust response to refeeding, RSC activity must be poised to be active in this state. An intriguing possibility could be that histone acetylation inhibits RSC activity to some extent as was recently reported in vitro (*Lorch et al., 2018*). This would be consistent with the rapid changes in nucleosome positioning at many genes during quiescence exit in the absence of high levels of histone acetylation. (2) Recent structural studies have shown that the nucleosome acidic patch is in direct contact with subunits of the RSC complex (*Ye et al., 2019*; *Patel et al., 2019*; *Wagner et al., 2020*; *Baker et al., 2021*). If the acidic patch is occluded by hypoacetylated H4 tails in quiescence for example (*Luger et al., 1997*; *Shogren-Knaak et al., 2006*; *Robinson et al., 2008*; *Allahverdi et al., 2011*; *Swygert et al., 2020*), it is possible that RSC can no longer interact with this region of the nucleosome, rendering its binding abilities different in quiescence. Finally, (3) a lack of Pol II activity in quiescent cells could prevent RSC from moving out of NDRs and into gene bodies. Indeed, transcription appears to play a prominent role in RSC localization: RSC moves into gene bodies during transcription activation and this movement is blocked when transcription is inhibited, as we have reported above. It is likely that a combination of transcription and histone acetylation helps pull RSC into gene bodies, given recent work showing that acetylation is a consequence of transcription (*Martin et al., 2021*).

An additional model we favor is one in which RSC's activity is reduced in quiescence, in part, due to reduced ATP levels during glucose starvation (*Özalp et al., 2010*; *Laporte et al., 2011*; *Joyner et al., 2016*). It is possible, then, that we could infer RSC activity from its binding pattern at NDRs versus at the +1 nucleosome and beyond. According to this model, RSC sitting at NDRs in quiescence is inactive or has low biochemical activities. RSC-dependent chromatin remodeling could then be greatly aided by high levels of Pol II upon quiescence exit. Pol II is known to disrupt nucleosomes, which facilitates binding of other chromatin regulators (*Kireeva et al., 2005*; *Schwabish and Struhl, 2004*; *Kulaeva et al., 2010*; *Martin et al., 2018*). Nucleosome disruption by Pol II could thus allow RSC to function more readily in low ATP conditions during early stages of quiescence exit. Consistent with this model, we see high Pol II activity relative to other sites at a subset of genes in quiescent cells, where RSC localizes to fragile nucleosomes and outside the NDR at the +1 nucleosomes. Additionally, at these sites, RSC moves more readily toward gene bodies during quiescence exit as Pol II occupancy increased.

In a separate study, we recently found that the SWI/SNF remodeling enzyme promotes transcription of a subset of hypoacetylated genes during quiescence entry, implying a specialized transcription regulation program for essential genes in the wake of widespread transcriptional shutdown (*Spain et al., 2018*). In cycling cells, it was recently shown that RSC and SWI/SNF cooperate at a

subset of genes (*Rawal et al., 2018*). Our results suggested that cooperation between the two SWI/SNF class remodeling factors may also occur during quiescence entry.

Consistent with co-transcriptional re-localization, our data suggest that RSC plays an active role in helping Pol II transcribe past the +1 nucleosome in addition to initiating transcription. Supporting this idea was our observation of a subset of genes where RSC depletion caused a Pol II enrichment around the +1 nucleosome. Previous reports showed that RSC can bind gene bodies and impact elongating and terminating Pol II (*Spain et al., 2014*; *Ocampo et al., 2019*), and one study showed interactions between the Rsc4 subunit and all three RNA polymerases (*Soutourina et al., 2006*). An intriguing possibility could be that RSC directly interacts with Pol II to facilitate transcription past the first few nucleosomes.

The transcriptional response during quiescent exit was dampened by depleting the essential chromatin remodeler, RSC, but it did not diminish completely. Pol II occupancy was globally decreased approximately twofold at the 10 min time point in RSC-depleted cells. However, in some cases, we found reduced sense transcription and increased antisense transcription. This was largely due to a nearby NDR susceptible to transcription initiation that could be co-opted for antisense transcription. The mechanism that allows for this cryptic transcription is still unknown. Chromatin-remodeling enzymes are vastly important for repressing antisense lncRNAs (*Han and Chang, 2015*). Different chromatin-remodeling enzymes function to repress lncRNA transcripts in cycling cells, including RSC (*Alcid and Tsukiyama, 2014*; *Marquardt et al., 2014*; *Gill et al., 2020*). We speculate RSC is particularly suitable to regulate global transcriptome during quiescence exit due to its high abundance, which allows it to function through multiple mechanisms. The mouse embryonic stem-cell-specific BAF complex was also recently shown to globally repress lncRNA expression (*Hainer et al., 2015*). This raises the possibility that some of our observations in yeast quiescent cells could be conserved in mammalian quiescent cells. Given the robust transcriptional response that occurs during quiescence exit, it is likely that chromatin structure is crucial for maintaining the quality of the transcriptome. Indeed, we noted cases where transcription occurred upstream of the canonical TSS when an NDR was not generated, highlighting the defects in Pol II initiation and start site selection due to chromatin defects in the absence of RSC. Hypertranscription events similar to the one observed during quiescence exit occur throughout all organisms, particularly during development (*Percharde et al., 2017*). Therefore, it is quite possible that we will see similar, multifaceted roles for RSC homologs or other abundant chromatin-remodeling factors in facilitating proper hypertranscription in many other systems.

# Materials and methods

## Key resources table

| Reagent type (species) or resource | Designation | Source or reference | Identifiers | Additional information |
|---|---|---|---|---|
| Strain, strain background (*S. cerevisiae*) | WT; prototroph | Tsukiyama Lab | YTT5781 | *MATa RAD5+* |
| Strain, strain background (*S. cerevisiae*) | WT; prototroph | Tsukiyama Lab | YTT5782 | *MATα RAD5+* |
| Strain, strain background (*S. cerevisiae*) | Sth1 and Sfh1 degrons | Tsukiyama Lab | YTT7222 | *MATa can1-100 RAD5 + Sth1-3HSV-IAA1-T10-KanMX Sfh1-3HSV-IAA1-T10-Hyg* |
| Strain, strain background (*S. cerevisiae*) | Sth1 and Sfh1 degrons | Tsukiyama Lab | YTT7224 | *MATα can1-100 RAD5 + Sth1-3HSV-IAA1-T10-KanMX Sfh1-3HSV-IAA1-T10-Hyg* |
| Strain, strain background (*S. cerevisiae*) | *dst1Δ* | Tsukiyama Lab | YTT7308 | *MATa RAD5 + dst1Δ::KanMX* |

*Continued on next page*

*Continued*

| Reagent type (species) or resource | Designation | Source or reference | Identifiers | Additional information |
|---|---|---|---|---|
| Strain, strain background (*S. cerevisiae*) | *dst1Δ* | Tsukiyama Lab | YTT7309 | *MATa RAD5 + dst1Δ::KanMX* |
| Chemical compound, drug | Indole-3-acetic acid (IAA) | Sigma | I3750-5G-A | 1 mg/mL powder added to culture |
| Antibody | Rpb3 (mouse monoclonal) | Biolegend | 665003 | WB: (1:1000) dilution ChIP: (2 µL) |
| Antibody | Ser5p (rat monoclonal) | Active Motif | 61085 | WB: (1:1000) |
| Antibody | Ser2p (rat monoclonal) | Active Motif | 61083 | WB: (1:1000) |
| Antibody | HSV (rabbit polyclonal) | Sigma | H6030-200UG | WB: (1:5000) |
| Antibody | H3 (rabbit polyclonal) | Abcam | 1791 | WB: (1:1000) ChIP: (1 µL) |
| Antibody | Flag (mouse monoclonal) | Sigma | F1804 | ChIP: (2 µL) |
| Other | Protein G magnetic | Invitrogen | 10004D | ChIP: (20 µL) |
| Peptide, recombinant protein | Zymolyase 100T | AMSBIO | 120493–1 | MNase-seq; 10 mg per 100 units $OD_{660}$ cells |
| Peptide, recombinant protein | Micrococcal nuclease | Worthington | LS004798 | MNase-seq 50U (high digests) 5U (low digests) |
| Other | AMPure XP | Beckman | A63880 | - |
| Strain, strain background (*K. lactis*) | Spike-in control strain | Nathan Clark Lab | NRRL Y-1140 | 100:1 cell mixture (*S. cerevisiae*: *K. lactis*) |
| Chemical compound, drug | 4-Thiouracil | Sigma | 440736–1G | 5 mM |
| Commercial assay or kit | RiboPure Yeast Kit | Thermo Fisher | AM1926 | |
| Chemical compound, drug | MTSEA biotin-XX | Biotium | 90066 | 16.4 µM in 20 mM HEPES pH 7.4 1 mM EDTA |
| Other | Streptavidin beads | Invitrogen | 65001 | (40 µL) |
| Commercial assay or kit | miRNeasy kit | Qiagen | 217084 | - |
| Commercial assay or kit | Ovation SoLo kit; custom AnyDeplete | NuGEN/Tecan | Contact rep for custom reagent (yeast rRNA depletion) | - |

## Yeast strains, yeast growth media, quiescent cell purification, and exit time courses

The *S. cerevisiae* strains used in this study are listed in the Key Resources Table and are isogenic to the strain W303-1a with a correction for the mutant *rad5* allele in the original W303-1a (*Thomas and Rothstein, 1989*). Yeast transformations were performed as previously described (*Ausubel FM et al., 1988*). All cells were grown in YPD medium (2% Bacto Peptone, 1% yeast extract, 2% glucose). We note that quiescent (Q) yeast need to be grown in YPD using 'fresh' (within ~3 months) yeast extract as a source. To purify Q cells, liquid YPD cultures were inoculated with a single colony into liquid cultures (colonies were no older than 1 week). Yeast cells were grown in Erlenmeyer flasks

10 times the liquid volume for 7 days at 30°C and shaking at 180 RPM. Q cells were purified by percoll gradient centrifugation as previously described (*Allen et al., 2006*). Briefly, percoll was diluted 9:1 with 1.5 M NaCl into 25 mL Kimble tubes and centrifuged at 10,000 RPM for 15 min at 4°C. Seven-day cultures were pelleted, washed with ddH$_2$O, resuspended in 1 mL of ddH$_2$O, and gently pipetted over a pre-mixed percoll gradient. Four hundred OD$_{660}$ were pipetted onto a 25 mL gradient. Gradients with loaded cells were centrifuged for one hour at 1000 RPM, 4°C. The upper, non-quiescent cell population and the middle, ~8 mL fraction, were carefully discarded via pipetting. The remaining volume was washed twice with ddH$_2$O in a 50 mL conical tube at 3000 RPM, 10 min each.

Q exit experiments were performed as follows: Q cells were harvested and added to YPD to 1 OD$_{660}$/mL. Cells were grown at 25°C to slow the kinetics for feasibility. For ChIP-seq and MNase-seq experiments, cells were grown to the appropriate time and then crosslinked for 20 min (described in more detail in the sections below).

## Depletion of RSC subunits, Sth1 and Sfh1

The yeast strains YTT 7222 and 7224 were grown in 5 mL overnight YPD cultures, back diluted for four doublings, and inoculated to 0.002 OD$_{660}$ into the appropriate YPD volume for a given experiment. Cells were grown for 16 hr and monitored for glucose exhaustion using glucose strips. Six hours after glucose exhaustion, 1 mg/mL of indole-3-acetic acid (IAA) (Sigma, I3750-5G-A) was added, in powder form, to the culture. IAA remained in the culture for 7 days before harvesting Q cells. Q cells were purified as described above, and depletion efficiency was determined by western blot analysis (*Figure 3—figure supplement 1B*).

## Western blot analysis

Yeast cells were lysed by bead beating in trichloroacetic acid, as previously described (*Cox et al., 1997*). Proteins were resolved on 8% polyacrylamide gels and transferred to nitrocellulose membranes. Membranes were incubated with primary antibodies: anti-Rpb3 (Biolegend, 665003 1:1000 dilution), anti-Ser5p (Active Motif, 61085 1:1000 dilution), anti-Ser2p (Active Motif, 61083, 1:1000 dilution), and anti-HSV (Sigma, 1:500). Following primary incubation, membranes were incubated with either anti-mouse or anti-rabbit secondary antibodies (Licor, 1:10,000). Protein signals were visualized by the Odyssey CLx scanner.

## ChIP-seq

One hundred OD$_{660}$ U of cells were crosslinked and sonicated in biological duplicate using the protocol described in *Rodriguez et al., 2014*. Proteins were immunoprecipated from 1 µg chromatin and 1 µL of anti-H3 (Abcam, 1791) conjugated to 20 µL protein G magnetic beads (Invitrogen, 10004D) per reaction. For Pol II ChIPs, we used an antibody against the Rpb3 subunit (2 µL per reaction, Biolegend 665004) conjugated to 20 µL protein G magnetic beads (Invitrogen, 10004D). For Sth1 ChIP experiments, we used an antibody against the Flag-epitope tag, FLAG M2 mouse monoclonal (Sigma Aldrich, F1804) and conjugated to 20 µL protein G beads (Invitrogen, 10004D). Libraries were generated using the Ovation Ultralow v2 kit (NuGEN/Tecan, 0344) and subjected to 50 bp single-end sequencing on an Illumina HiSeq 2500 at the Fred Hutchinson Cancer Research Center genomics facility.

## ChIP-seq analysis

We used bowtie2 to align raw reads to the sacCer3 reference genome (*Langmead and Salzberg, 2012*). Reads were then filtered using SAMtools (*Li et al., 2009*). Bigwig files of input-normalized ChIP-seq data were generated from the filtered bam files using deepTools2 (*Ramírez et al., 2014*) and dividing the IP data by the input data. All ChIP-seq IP data were normalized to RPKM and the corresponding input samples. Pol II ChIP-seq data were both input normalized and spike-in normalized. Matrices for metaplots were generated in deepTools2 using the annotation file from *Xu et al., 2017*. Clustering was performed using the k-means function in deepTools2. For GO analysis, the lists of genes within each cluster were entered into http://geneontology.org/ database and the first five GO terms with an false discovery rate of <0.05 are shown in *Figure 1—figure supplement 2*.

## MNase-seq

Cell growth and crosslinking were done in the same fashion as in ChIP-seq experiments. Generally, we followed the protocol in *Rodriguez et al., 2014*, with changes described here. Cells were spheroplasted using 10 mg zymolyase (100T, AMSBIO, 120493–1) per 100 OD$_{660}$ cells. For Q cells, zymolyase treatment could take up to 2 hr. We monitored the cells via microscopy and stopped the spheroplasting step when ~80% of the cells were spheroplasted. MNase digestion was performed as described in *Rodriguez et al., 2014*. High digests (80% mononucleosomes) required 50 U of MNase (Worthington, LS004798), and for the low digests, chromatin was treated with 10 U of MNase. From this step, chromatin was reverse crosslinked as described in *Rodriguez et al., 2014*. Following reverse crosslinking, RNase, and proteinase-K digestion, DNA was phenochloroform extracted. Any large, uncut genomic DNA species was separated out using Ampure beads (Beckman). Sequencing libraries were generated from the purified DNA using the Ovation Ultralow v2 kit (NuGEN, 0344). Libraries were subjected to 50 bp paired-end sequencing on an Illumina HiSeq 2500 at the Fred Hutchinson Cancer Research Center genomics facility.

## MNase-seq analysis

We used bowtie2 to align raw reads to the sacCer3 genome and filtered reads using SAMtools as described above for ChIP-seq analysis. Bigwig files of input-normalized ChIP-seq data were similarly generated from the filtered bam files using deepTools2 and the MNase option to center the reads around nucleosome dyads. Data represented in the paper were filtered to mononucleosome sizes using deepTools2. Mapped reads were normalized by RPKM. For NDR-width quartiles shown in *Figure 3*, NDRs were sorted into decreasing width and then divided by four. Each cluster is 25% of the NDRs.

## Nascent RNA-seq

Generally, nascent RNA-seq experiments were performed as described in *Bonnet et al., 2014*; *Duffy et al., 2015*. For the 0 min and 5 min samples, we added 100 and 50 OD$_{660}$ of Q cells, respectively, to YPD containing 5 mM 4-thiouracil (Sigma, 440736–1G). Cells were incubated with 4tU for 5 min before pelleting (1 min, 3500 RPM) and flash frozen in liquid nitrogen. For the 10 min time points, 50 OD units of quiescent cells were released into YPD for 5 min before an additional 5 min incubation with 4tU at a final concentration of 5 mM. All time points were labeled with 4tU for a total of 5 min before pelleting and freezing. Total RNA was isolated using Ambion's RiboPure Yeast Kit (Thermo, AM1926). *S. cerevisiae* cells were lysed in the presence of *Kluvomyces lactis* (*K. lactis*) cells in a 100:1 mixture. RNA was treated with DNAseI according to the TURBO DNase kit (Thermo, AM2238). Forty microgram RNA was then biotinylated with MTSEA biotin-XX (diluted in 20% DMF) at a final concentration of 16.4 µM in 20 mM HEPES pH 7.4 and 1 mM EDTA at room temperature for 30 min. Unreacted MTS-biotin was removed from samples by phenol:chloroform:isoamyl-alcohol extraction and resuspended in 100 µL nuclease-free water. Strepavidin beads (Invitrogen 65001) were washed with high-salt wash buffer (100 mM Tris, 10 mM EDTA, 1 M NaCl, 0.05% Tween-20) and blocked for 1 hr in high-salt wash buffer containing 40 ng/µL glycogen. Forty microliters of streptavidin beads were added to the RNA samples and incubated for 15 min at room temperature. Beads were washed three times in 1 mL high-salt wash buffer and eluted for 15 min at room temperature in 50 µL streptavidin elution buffer (100 mM DTT, 20 mM HEPES, 2.7, 1 mM EDTA, 100 mM NaCl, 0.05% Tween-20). The resulting RNA was then purified and concentrated using the Qiagen miRNeasy kit (#217084). Libraries were prepared from 5 ng of RNA using the Ovation SoLo kit (NuGEN/Tecan, custom AnyDeplete; contact Tecan for ordering this kit for yeast). Libraries were subjected to 50 bp paired-end sequencing on an Illumina HiSeq 2500 at the Fred Hutchinson Cancer Research Center genomics facility.

## Nascent RNA-seq analysis

We used bowtie2 to align raw reads to the sacCer3 and *K. lactis* (Ensembl ASM251v1) genomes and filtered reads using SAMtools as described above for ChIP-seq analysis. Reads were normalized to the spike-in control and RPKM. Differential expression analysis was performed using DESeq2 (*Love et al., 2014*). For *Figure 6*, sense transcripts from log2 ratio data (−RSC/+RSC) were sorted into three k-means clusters. The cluster containing enriched upstream transcripts was used for

further analysis and is shown in *Figure 6*. Clustering information is also provided in the source data files.

## Acknowledgements

We are grateful to members of the Tsukiyama lab and Pravrutha Raman for helpful comments and critical reading of this manuscript. We thank Sarah Hainer and Felix Mueller-Planitz for advice on MNase-seq experiments. We thank Benjamin Martin, Rafal Donczew, and Sandipan Brahma for advice and feedback. We thank Mitchell Ellison and Alex Francette for advice about analyzing nascent RNA-seq data. TT was supported by the National Institutes of Health (R01 GM111428 and R35GM139429). CEC was supported by the National Cancer Institute (T32CA009657) and National Institutes of Health (F32GM131554).

## Additional information

### Funding

| Funder | Grant reference number | Author |
| --- | --- | --- |
| NCI | T32CA009657 | Christine E Cucinotta |
| NIGMS | F32GM131554 | Christine E Cucinotta |
| NIGMS | R01 GM111428 | Toshio Tsukiyama |
| NIGMS | R35GM139429 | Toshio Tsukiyama |

The funders had no role in study design, data collection and interpretation, or the decision to submit the work for publication.

### Author contributions

Christine E Cucinotta, Conceptualization, Data curation, Formal analysis, Funding acquisition, Investigation, Methodology, Writing - original draft, Writing - review and editing; Rachel H Dell, Keean CA Braceros, Investigation; Toshio Tsukiyama, Conceptualization, Supervision, Funding acquisition, Investigation, Project administration, Writing - review and editing

### Author ORCIDs

Christine E Cucinotta (iD) https://orcid.org/0000-0002-9644-3126
Toshio Tsukiyama (iD) https://orcid.org/0000-0001-6478-6207

### Decision letter and Author response

Decision letter https://doi.org/10.7554/eLife.67033.sa1
Author response https://doi.org/10.7554/eLife.67033.sa2

## Additional files

### Supplementary files

• Transparent reporting form

### Data availability

All sequencing data are uploading on the NCBI Gene Expression Omnibus under the accession number GSE166789.

The following dataset was generated:

| Author(s) | Year | Dataset title | Dataset URL | Database and Identifier |
| --- | --- | --- | --- | --- |
| Cucinotta CE, Dell RH, Braceros KC, Tsukiyama T | 2021 | RSC primes the quiescent genome for hypertranscription upon cell cycle re-entry | https://www.ncbi.nlm. nih.gov/geo/query/acc. cgi?acc=GSE166789 | NCBI Gene Expression Omnibus, GSE166789 |

The following previously published dataset was used:

| Author(s) | Year | Dataset title | Dataset URL | Database and Identifier |
|---|---|---|---|---|
| McKnight JN, Boerma JW, Breeden LL, Tsukiyama T | 2015 | Global Promoter Targeting of a Conserved Lysine Deacetylase for Transcriptional Shutoff during Quiescence Entry | https://www.ncbi.nlm. nih.gov/geo/query/acc. cgi?acc=GSE67151 | NCBI Gene Expression Omnibus, GSE67151 |

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
