## [Decision Letter]

**Acceptance summary:**

Cucinotta et al. describe the transcription that occurs within minutes of nutrient repletion as *Saccharomyces cerevisiae* yeast cells emerge from quiescence, noting an unexpected, intense burst of expression from much of the genome. They focus primarily on the role of the nucleosome remodeler RSC in managing chromatin architecture over promoters during quiescence and as cells re-enter the cell cycle using a broad range of genome-wide measurements that strongly support the conclusions. This important process of cell cycle re-entry from the quiescent state is understudied but impacts a range of phenomena from development and carcinogenesis in multicellular organisms to adaptation of microorganisms to environmental cues, so the results will be of interest to a broad audience.

**Decision letter after peer review:**

Thank you for submitting your article "RSC primes the quiescent genome for hypertranscription upon cell cycle re-entry" for consideration by *eLife*. Your article has been reviewed by 3 peer reviewers, including Tim Formosa as the Reviewing Editor and Reviewer #1, and the evaluation has been overseen by Jessica Tyler as the Senior Editor.

Essential revisions:

Note that while the reviewers felt strongly that more integration of, and cross-correlation among the datasets would improve the impact of this work, the suggestions for ways to do that listed below are individually optional. The long list of recommendations may look daunting, but the authors are encouraged to view them as potential options, not as a list that has to be completed.

1. Integration and correlation of different datasets should be enhanced. For example, a heatmap of the MNase-seq, TFIIB-seq, and RSC ChIP-seq data using the gene clusters in Figure 1C could be used to correlate induction of transcription with RSC occupancy, favorable NDR width, and binding of TFIIB. Another example would be to examine whether genes with transcription defects due to RSC depletion show greater changes in promoter nucleosome occupancy when compared to unaffected genes. Some attempt at this was made in Figures 6 and 7, but more analysis of this type would provide better support for the key conclusions.

2. Improvements should be made to the bioinformatic and statistical handling of the data. The existing bioinformatic tools employed are appropriate, but they are not consistently applied for different experiments. How specific gene groups were chosen or clustered should be more carefully explained, and, if appropriate, cut-offs listed. Similar information should be included to support statements in the text (i.e. ~1000 genes have fragile nucleosomes, Sth1 bound to ~80% of NDRs). Some of the data is "normalized" but it isn't always clear what it is normalized to (spike-ins? Input?). Statistical analyses should be used to back up key results.

3. Authors should acknowledge the impact of the delayed exit from quiescence following RSC depletion on their results. Cells that were able to exit are likely those with subtle differences in chromatin packing over a handful of genes required for exit. Cells that do not exit are left with quiescence signatures, which could be due to the lack of transcription as opposed to lack of RSC.

4. The authors Pol II occupancy data via ChIP-seq with and without RSC depletion. Could these locations be compared with the transcription changes shown in Figure 1 to demonstrate which groups of genes are altered?

5. In Figure 1, it is interesting that some of the clusters show sustained expression (Cluster 1), but some have hyper-transcription that is then reduced to log levels (Clusters 2 and 3). Is there a difference between these and RSC occupancy; or specific changes to one of these clusters when RSC is depleted? Furthermore, what genes make up these clusters?

6. In Figure 3a, the authors show a decrease in fragile nucleosome occupancy through low MNase upon exit from quiescence. Is the effect more pronounced in a subset of genes such as just quartile 1, with the widest NDR? Similarly, does an analysis of MNase data at only those locations with the highest RSC occupancy produce a different outcome?

7. Do the authors see decreased H3 occupancy (from ChIP-seq data) at the genes where the fragile nucleosome decreases (via MNase-seq data)?

8. In Figure 4, the authors find that Pol II occupancy never reaches the 10-minute hyper-transcription enrichment when RSC is depleted. Do the cells ever reach this peak, or are they unable to obtain this hyper-transcription in the absence of RSC?

9. Figure 5A shows that RSC occupancy changes to decrease over NDRs and increase over gene bodies. The authors then follow with inhibiting Pol II, and show an accumulation of Pol II at NDRs and decreasing over gene bodies. Do these data correlate? For example, does an ROC curve show correlation? Or if you compare the datasets do they show that the largest changes in Pol II distribution occur at the locations where RSC redistributes?

10. Does the increased aberrant transcription shown in Figure 7 correlate with the level of RSC occupancy or the change in occupancy of Pol II?

11. While Figure 8 accurately summarizes the key results of this study, more molecular insight either here or in the discussion would be helpful. One model is that RSC has a preference for "easy to remodel" nucleosomes, such as fragile nucleosomes. In quiescence, due to lack of ATP, RSC might be less able to evict these nucleosomes and becomes stuck at NDRs. Following the eviction of fragile nucleosomes at cells exit quiescence, RSC might move onto other "easy to remodel" nucleosomes, such as those partially disrupted by Pol II. This, or some mechanism preferred by the authors, might be discussed to help the reader.

*Reviewer #1 (Recommendations for the authors):*

1) In the abstract, the authors state that ~60% of the genome is transcribed during the first 10 minutes after exit, but the data in figure 1 state that ~20% of genes are transcribed above the cutoff. It would be useful to reconcile this apparent discrepancy, and also to state whether the percentage of transcription is based on gene number or the total fraction of the genome.

2) On line 156, the authors note a result in "Fig. 2-supplement 1B" but there is no figure that matches this description or seems to be related to the statement being supported.

3) On line 169, the authors state that "the MNase-sensitivity of the +1 and +2 nucleosomes increased at the 10-minute time point" but this conclusion is not evident in Fig 3. The effect on the left side of the NDR in 3A-3B is clear, but the reader will need more guidance with seeing the change that supports the statement above.

4) The flow cytometry result in Fig 4A +RSC seems like it should be similar to the results in Fig1-Sup-D and 2F, but instead it looks like a fully occupied G2 delay/arrest. Can the authors explain the difference?

5) In Fig 5C, the authors describe the rise in Pol II ChIP signal at +1 Kb as a termination delay. It seems more appropriate to perform gene-averaging (TSS to TTS) or to align all termination sites instead of looking at 1 Kb past the TSS, which is early in some genes and late in others.

6) The basis for placing genes in Class I and II in Fig 7 is not clear. The text says that Class I has high antisense and "generally reduced sense" transcripts. This seems mostly true, although about 20% of the Sense levels are up significantly. Class II, however, is described as having "modest changes in both sense and antisense transcript levels," but sense ranges from significantly up to significantly down, and antisense from down to strongly down. The authors should describe the rationale for defining these classes more clearly; are they based initially on levels of antisense transcripts? Without a clearer understanding of how these classes are defined, it is difficult to determine whether or not the conclusions drawn are supported.

7) On line 327, it is not clear what an "incredibly strong" transcriptional response would be, so another adjective might be more appropriate.

*Reviewer #2 (Recommendations for the authors):*

I find the data very compelling and complete in the description of RSC regulating exist from quiescence. As mentioned in the public review, some aspects of data comparisons could be made. Some examples are listed below. While some may not be necessary, some of these may help to strengthen conclusions:

1) The authors Pol II occupancy data via ChIP-seq with and without RSC depletion. Could these locations be compared with the transcription changes shown in Figure 1 to demonstrate which groups of genes are altered?

2) In Figure 1, it is interesting that some of the clusters show sustained expression (Cluster 1), but some have hypertranscription that is then reduced to log levels (Clusters 2 and 3). Is there a difference between these and RSC occupancy; or specific changes to one of these clusters when RSC is depleted? Furthermore, what genes make up these clusters?

3) In Figure 3a, the authors show a decrease in fragile nucleosome occupancy through low MNase upon exit from Q. This is a metaplot of all genes, and I am curious if the authors see more pronounced results if they show just quartile 1, with the widest NDR. Also, if the authors analyze the MNase data and show only locations that have RSC occupancy.

4) Do the authors see decreased H3 occupancy (from ChIP-seq data) at the genes where the fragile nucleosome decreases (via MNase-seq data)?

5) In Figure 4, the authors find that Pol II occupancy never reaches the 10-minute hypertranscription enrichment when RSC is depleted. Curiously, do the cells ever reach this peak, or are they unable to obtain this hypertranscription in the absence of RSC?

6) Figure 5A shows that RSC occupancy changes to decrease over NDR and increase over gene bodies. The authors then follow with inhibiting Pol II, and show an accumulation of Pol II at NDRs and decreasing over gene bodies. Do these data correlate? For example, does an ROC curve show correlation? Or if you compare the datasets do they show that at the locations where RSC redistributes, there is the largest change in Pol II distribution?

7) The locations that have increased aberrant transcription, as shown in Figure 7, are these genes that have RSC occupancy? If you compare these locations with Pol II altered occupancy, is there a correlation?

*Reviewer #3 (Recommendations for the authors):*

1. It is surprising that little attempt was made to integrate the results from the various analyses. For example, a heatmap of the MNase-seq, TFIIB-seq, and RSC ChIP-seq data using the gene clusters in Figure 1C would show that gene induction following Q exit is linked to RSC occupancy, favorable NDR width, and binding of TFIIB. Another example would be to examine whether genes with transcription defects due to RSC depletion show greater changes in promoter nucleosome occupancy when compared to unaffected genes. Some attempt at this was made in Figures 6 and 7, but I thought that more connections would better support the key results of this study.

2. Improvements should be made to the bioinformatic and statistical handling of the data. The existing bioinformatic tools employed are fine, but they are not consistently applied for different experiments. How specific gene groups were chosen or clustered should be more carefully explained, and, if appropriate, cut-offs listed. Similar information should be included to support statements in the text (i.e. ~1000 genes have fragile nucleosomes, Sth1 bound to ~80% of NDRs). Some of the data is "normalized" but it isn't always clear what it is normalized to (spike-ins? Input?). Statistical analyses should be used to back up key results.

3. Authors should acknowledge the impact of the delayed Q-exit following RSC depletion on their results. Cells that were able to exit are likely those with subtle differences in chromatin packing over a handful of genes required for Q exit (down the road identifying these might be possible using SGA?). Cells that do not exit are left with Q signatures, which could be due to the lack of transcription upon Q exit as opposed to lack of RSC.

4. While Figure 8 accurately summarizes the key results of this study, more molecular insight either here or in the discussion would be helpful. One model is that RSC has a preference for "easy to remodel" nucleosomes, such as fragile nucleosomes. In Q, due to lack of ATP, RSC cannot evict these nucleosomes and becomes stuck at NDRs. Following the eviction of fragile nucleosomes at Q-exit, RSC moves onto other "easy to remodel" nucleosomes, such as those partially disrupted by PolII.

---

## [Author Response]

Essential revisions:Note that while the reviewers felt strongly that more integration of, and cross-correlation among the datasets would improve the impact of this work, the suggestions for ways to do that listed below are individually optional. The long list of recommendations may look daunting, but the authors are encouraged to view them as potential options, not as a list that has to be completed.1. Integration and correlation of different datasets should be enhanced. For example, a heatmap of the MNase-seq, TFIIB-seq, and RSC ChIP-seq data using the gene clusters in Figure 1C could be used to correlate induction of transcription with RSC occupancy, favorable NDR width, and binding of TFIIB. Another example would be to examine whether genes with transcription defects due to RSC depletion show greater changes in promoter nucleosome occupancy when compared to unaffected genes. Some attempt at this was made in Figures 6 and 7, but more analysis of this type would provide better support for the key conclusions.

We are very grateful for this suggestion and have added several panels addressing these questions to a new figure: Figure 4—supplement 1. In summary, we found the following: genes with widest NDRs (clusters 1 and 2) have high levels of TFIIB (Figure 4—supplement 1A) and Pol II during early exit. While these two clusters were impacted by RSC depletion, some level of Pol II occupancy remained (Figure 4—supplement 1B). We believe redundant factors may enable transcription to occur at these genes (ex: other remodelers or transcription factors). Furthermore, these genes (clusters 1 and 2) also had the widest NDRs and lowest H3 occupancy regardless of RSC, although we did see H3 occupancy rise in the absence of RSC (Figure 4—supplement 4C). RSC occupancy levels did not correlate with transcription levels or NDR width (highest RSC occupancy present at clusters 2 and 4; Figure 4—supplement 4D), with cluster 1 transcription and NDR width deviating from RSC occupancy the most. We again believe this could be due to redundant factors at loci in cluster 1.

2. Improvements should be made to the bioinformatic and statistical handling of the data. The existing bioinformatic tools employed are appropriate, but they are not consistently applied for different experiments. How specific gene groups were chosen or clustered should be more carefully explained, and, if appropriate, cut-offs listed. Similar information should be included to support statements in the text (i.e. ~1000 genes have fragile nucleosomes, Sth1 bound to ~80% of NDRs). Some of the data is "normalized" but it isn't always clear what it is normalized to (spike-ins? Input?). Statistical analyses should be used to back up key results.

We thank the reviewers for this feedback and have updated the text to better describe how the analyses were completed, for example page 26, line 683. In addition, we added a section for analysis for each experiment type in the methods section.

3. Authors should acknowledge the impact of the delayed exit from quiescence following RSC depletion on their results. Cells that were able to exit are likely those with subtle differences in chromatin packing over a handful of genes required for exit. Cells that do not exit are left with quiescence signatures, which could be due to the lack of transcription as opposed to lack of RSC.

We agree this should be acknowledged and we have updated the text (page 12, lines 273-275) to discuss this possibility.

4. The authors Pol II occupancy data via ChIP-seq with and without RSC depletion. Could these locations be compared with the transcription changes shown in Figure 1 to demonstrate which groups of genes are altered?

We have added this analysis using the same gene ordering as shown in Figure 1 and generated a new supplemental figure (Figure 4—supplement 1) to illustrate this comparison. These results are now discussed in page 12, lines 262-268.

5. In Figure 1, it is interesting that some of the clusters show sustained expression (Cluster 1), but some have hyper-transcription that is then reduced to log levels (Clusters 2 and 3). Is there a difference between these and RSC occupancy; or specific changes to one of these clusters when RSC is depleted? Furthermore, what genes make up these clusters?

We agree that the differences in gene expression varied over time across the clusters were interesting. We have added a heatmap of RSC occupancy in Q cells and during early exit at these loci (new Figure 4—supplement 1), and discuss in page 12, lines 261-270. We note differences in the RSC profile: specifically striking to us was how much RSC remains in the NDR vs the gene body. Genes showing more RSC movement into the gene body (clusters 1 and 2) also had the highest binding of Pol II in exit. All genes had reduced Pol II occupancy however clusters 1 and 2 still had detectable levels of Pol II occupancy. We included a new figure showing the top five GO terms for each cluster (new Figure 1—supplement 2), and discuss on page 7, lines 129-131. Cluster 1 had the highest enrichment of GO terms and contained genes involved in rRNA processing and translation.

6. In Figure 3a, the authors show a decrease in fragile nucleosome occupancy through low MNase upon exit from quiescence. Is the effect more pronounced in a subset of genes such as just quartile 1, with the widest NDR?

We see a moderate effect of Q-exit on MNase-digestion ability after the data are subset into quartiles (shown in new Figure 3—supplement 1A, and discussed on page 9, lines 181-184).

Similarly, does an analysis of MNase data at only those locations with the highest RSC occupancy produce a different outcome?

We thank the reviewers for this question and have added this analysis to the paper. We see a difference in MNase-digestion levels comparing high RSC levels to the lowest RSC levels (New Figure 3—supplement 2, discussed in page 10, lines 208-209).

7. Do the authors see decreased H3 occupancy (from ChIP-seq data) at the genes where the fragile nucleosome decreases (via MNase-seq data)?

Yes. We see decreased H3 occupancy at these genes when cells exit. We show these metaplots in a new figure (New Figure 3—supplement 1, discussed in page 9, lines 184-186).

8. In Figure 4, the authors find that Pol II occupancy never reaches the 10-minute hyper-transcription enrichment when RSC is depleted. Do the cells ever reach this peak, or are they unable to obtain this hyper-transcription in the absence of RSC?

We have performed ChIP-seq analysis to the 30-minute time point and did not observe transcription levels reaching this peak. We have updated the text to emphasize that the level could potentially reach this later on (page 12, line 257).

9. Figure 5A shows that RSC occupancy changes to decrease over NDRs and increase over gene bodies. The authors then follow with inhibiting Pol II, and show an accumulation of Pol II at NDRs and decreasing over gene bodies. Do these data correlate? For example, does an ROC curve show correlation? Or if you compare the datasets do they show that the largest changes in Pol II distribution occur at the locations where RSC redistributes?

We have added a figure to reflect the clusters shown in Figure 1C (new version of Figure 5—supplement 1B). We believe that this illustrates the difference of RSC localization with and without phenanthroline, as there are further differences at different clusters. The most striking changes are in clusters 1 and 2, where Pol II is normally highly bound. At these sites, RSC movement is dramatically sequestered to the NDR when phenanthroline is added, supporting our conclusion that RSC co-transcriptionally moves into gene bodies. This result is now discussed in page 13, lines 294-297.

10. Does the increased aberrant transcription shown in Figure 7 correlate with the level of RSC occupancy or the change in occupancy of Pol II?

We thank the reviewers for this question. While it would have been very interesting if so, we did not observe any correlation with the levels of RSC occupancy or Pol II occupancy. We believe that increased aberrant transcription occurs when an unaffected or “wide-enough” NDR is present and increased transcription is thus an indirect effect of RSC depletion rather than direct action of RSC at these loci (discussed on page 16, lines 404-410).

11. While Figure 8 accurately summarizes the key results of this study, more molecular insight either here or in the discussion would be helpful. One model is that RSC has a preference for "easy to remodel" nucleosomes, such as fragile nucleosomes. In quiescence, due to lack of ATP, RSC might be less able to evict these nucleosomes and becomes stuck at NDRs. Following the eviction of fragile nucleosomes at cells exit quiescence, RSC might move onto other "easy to remodel" nucleosomes, such as those partially disrupted by Pol II. This, or some mechanism preferred by the authors, might be discussed to help the reader.

We are grateful to the reviewers for these suggestions and are in favor of this as a potential model. We have added these ideas to the discussion on page 19 line 479-491.